# MAF: An algorithm based on multi-agent characteristics for infrared and visible video fusion

**Yandong Liu**, **Linna Ji** *, **Fengbao Yang**, **Xiaoming Guo**

School of Information and Communication Engineering, North University of China, Taiyuan, China

* jlnnuc@163.com

## Abstract

Addressing the limitation of existing infrared and visible video fusion models, which fail to dynamically adjust fusion strategies based on video differences, often resulting in suboptimal or failed outcomes, we propose an infrared and visible video fusion algorithm that leverages the autonomous and flexible characteristics of multi-agent systems. First, we analyze the functional architecture of agents and the inherent properties of multi-agent systems to construct a multi-agent fusion model and corresponding fusion agents. Next, we identify regions of interest in each frame of the video sequence, focusing on frames that exhibit significant changes. The multi-agent fusion model then perceives the key distinguishing features between the images to be fused, deploys the appropriate fusion agents, and employs the effectiveness of fusion to infer and determine the fusion algorithms, rules, and parameters, ultimately selecting the optimal fusion strategy. Finally, in the context of a complex fusion process, the multi-agent fusion model performs the fusion task through the collaborative interaction of multiple fusion agents. This approach establishes a multi-layered, dynamically adaptable fusion model, enabling real-time adjustments to the fusion algorithm during the infrared and visible video fusion process. Experimental results demonstrate that our method outperforms existing approaches in preserving key targets in infrared videos and structural details in visible videos. Evaluation metrics indicate that the fusion outcomes obtained using our method achieve optimal values in 66.7% of cases, with sub-optimal and higher values accounting for 80.9%, significantly surpassing the performance of traditional single fusion methods.

## 1. Introduction

Infrared and visible video sensing technologies rely on distinct radiative physical properties and imaging methods to capture the key characteristics and differences of scene targets, demonstrating significant complementarity between the two video types. The fusion of infrared and visible video streams optimizes the imaging advantages of both, thereby enhancing comprehensive detection and accurate interpretation of video features [1]. Video fusion plays a critical role in various applications, including situational awareness in degraded visual environments, security monitoring and early warning in entrance areas, precise localization of

**Data availability statement:** All relevant data are within the paper and its Supporting Information files.

**Funding:** This work was supported by Basic Research Program of Shanxi Province (202203021221104). The funders had no role in study design, data collection and analysis, decision to publish, or preparation of the manuscript.

**Competing interests:** The authors have declared that no competing interests exist.

high-speed moving targets, and intelligent obstacle avoidance. It is particularly important in drone remote sensing, smart sensor-based driving, and hydraulic monitoring.

In recent years, researchers have increasingly recognized that extracting and utilizing image feature information to guide image fusion is an effective strategy for enhancing fusion quality and improving subsequent tasks. Lu et al [2]. introduced a triplet-based Siamese CNN for feature detection and extraction in thermal images. Subsequently, they presented SuperThermal [3], an end-to-end neural network designed for thermal feature learning. In our research, we propose a feature detection network inspired by the Faster-RCNN framework, integrating thermal enhancement and a multi-scale descriptor network to better capture feature representations in texture-less thermal images. Zhao et al[4]. developed a novel network for RGB-T semantic segmentation, effectively capturing contextual information. Nai et al [5]. proposed a dynamic feature fusion method based on spatiotemporal context, which deeply analyzes the characteristics of multiple visual features, leveraging the strengths of various features to address complex appearance changes and background clutter, thereby improving object tracking and detection performance.

Most fusion methods rely heavily on prior knowledge of the video, such as predefined fusion algorithms or pre-trained fusion models. These methods aim to identify efficient feature extraction techniques to capture source image features as comprehensively as possible, achieving satisfactory fusion results to some extent. However, the complexity and variability of video scenes make it difficult for predefined fusion models to meet the demands of continuously changing features. Consequently, methods that depend on specific transformations or representations have become a bottleneck in improving fusion quality. Pre-selected fusion algorithms may yield suboptimal results, and in extreme cases, may even cause fusion failure.

Dong et al[6–8]. proposed a fusion method that employs image feature information as a weight reference for the fusion strategy, allowing image feature information to fed back into the image fusion process. They introduced a novel fusion framework based on cybernetics, called FusionPID [9], which utilizes a proportional-integral-derivative (PID) control system to fuse infrared and visible images. FusionPID takes advantage of the control system's feedback capabilities for various fusion tasks, ensuring the fused image retains both the thermal radiation from the infrared image and the texture from the visible image. Later, they presented a collaborative fusion method for infrared and visible images based on pulse-coupled neural networks (PCNN) and PID control systems, called FusionCPP [10]. The controller adaptively adjusts the fusion weight according to the differences, enabling precise fusion of image features into the new image. These methods offer valuable insights for the multi-agent video fusion discussed in this paper.

Research indicates that the functional architecture of agents highly aligns with the video fusion process [11]. The deliberative agent [12], being the most intelligent type, can rigorously infer and reflect based on its understanding of the environment and its own intentions, making detailed action plans for collaboration and tasks. This aligns with the requirements of choosing optimal fusion strategies in video fusion. The self-learning agent updates its knowledge base in response to environmental changes [13], continually improving over time, consistent with the characteristics of storing fusion strategy knowledge bases during the fusion process. The reactive agent can respond quickly to external information [14], utilizing its rapid responsiveness to directly access fusion strategies from the existing knowledge base. Under a multi-scale fusion framework, the multi-agent fusion model achieves complex functions such as dynamic selection of fusion parameters, rules and algorithms through the collaboration of simple agents, while reducing system complexity and enhancing system robustness, reliability and flexibility.

In summary, agents autonomously perceive changes in the external environment and respond accordingly. By collaborating with other agents, they develop behavior strategies

suited for completing complex tasks, providing a theoretical framework for efficient fusion strategy selection. This paper addresses the limitation of existing fusion models, which are unable to dynamically adjust fusion strategies based on video differences, by proposing an infrared and visible video fusion algorithm that leverages multi-agent system characteristics. The algorithm constructs fusion agents with varying feature types, allowing agents to infer and determine the optimal fusion parameters, rules, and algorithms. Through inter-agent collaboration, the optimal fusion structure is achieved, overcoming the limitations of traditional models that cannot adapt fusion strategies dynamically.

The main contributions of this paper are as follows.

(1) Introducing the multi-agent system into the field of video fusion. Through the system's interactive collaboration mechanism, it can maintain balanced fusion results in the fused images when faced with different fusion tasks, improving the adaptability of the fusion method.

(2) Constructing a multi-agent video fusion model based on difference features, and designing perception modules, fusion agents, and decision components.

(3) Compared with existing fusion methods, our fusion model can achieve better fusion results and dynamically adjust according to the difference features within the video frames.

The structure of the remainder of this paper is as follows: Section 2 reviews the development of image fusion, provides background information, and discusses related work on multi-agent system theory. Section 3 presents the multi-agent video fusion model. Experimental results and discussions are provided in Section 4, while the conclusion is presented in Section 5.

## 2. Related work

### 2.1. Image fusion method

Nowadays, methods for infrared and visible fusion are broadly categorized into traditional techniques and those employing deep learning[15].Multi-scale transformation methods[16–19] generally employ transformation tools like Laplacian and discrete wavelets to break down source images into multi-scale coefficients, which are fused and then inversely transformed to achieve the final outcome. Although these methods perform well, their fusion effectiveness is influenced by the fixed structure of the algorithms. Subspace-based methods[20, 21] map high-dimensional features contained in input data to lower-dimensional features, utilizing techniques such as principal component analysis and independent component analysis. Traditional methods typically exhibit the following issues. (1) The reliance on manually designed conventional theories leads to complex algorithmic frameworks, and this method of using fixed mathematical transformations to extract features neglects the modal differences between source images. (2) The limited choice of fusion rules also restricts their performance to some extent.

Deep learning mainly involves extracting image features based on training sample data, allowing for the acquisition of deep, specific feature representations that enhance the efficiency and accuracy of image representation. For instance, Tang et al.[22] used ground truth labels for scene segmentation in their training, proposing SeAFusion. They constructed a scene segmentation network with semantic loss to predict the fusion image segmentation outcomes, utilizing semantic loss backpropagation to train the fusion network and enrich the semantic information of the fusion image. Although some methods have been adopted in place of ground truth labels, significantly enhancing the quality of fusion images, the potential

performance of CNNs cannot be fully realized without standard ground truth labels. Existing methods typically use convolution operations to extract features, capturing local characteristics of images; however, they tend to overlook long-distance dependencies within images, resulting in the loss of some global information during training. Chang et al.[23] developed an adaptive encoder-decoder network based on global Transformers, known as AFT. Zhou et al.[24] introduced a semantically supervised dual discriminator generative adversarial network (SDDGAN). By designing an Information Quantity Discriminator (IQD), the features of semantic objects are maintained and the fusion process is supervised; a dual discriminator is used to discern the distribution of infrared and visible information in fusion images, preserving and enhancing their modal features, thus obtaining fusion images with high-level semantic awareness.

However, whether using traditional or deep learning-based methods, the decision-making in fusion methods primarily relies on prior knowledge of the video, such as predetermined fusion algorithms or trained fusion models. In practical applications, due to the complexity and variability of video scenes, a fixed fusion model cannot meet the requirements for dynamic fusion.

## 2.2. Relevant concepts of multi-agent systems

With the advancement of artificial intelligence, the application of swarm intelligence is becoming increasingly important in many scenarios. Utilizing the competition or cooperation of multiple agents to tackle complex real-world tasks and enhance system collaboration efficiency, and solving tasks that single agents cannot, it holds broad application prospects. Examples include traffic control, resource scheduling, competitive gaming, industrial robot control, autonomous driving, and drone coordination. Huang et al.[25] have applied multi-agent decision-making to the control of traffic network signals, treating cars and traffic lights as agents, and formulating strategies for each. By utilizing a hierarchical architecture for the agents, they achieved refined control, thereby enhance traffic efficiency. Bu et al.[26] have based their work on multi-agent scheduling and resource allocation (MA-SRA) to address delivery delays and errors in logistics SC management, enhancing operational efficiency and bringing significant advantages to the industry in terms of improved resource allocation, connectivity, delivery efficiency, and reduced delays and scheduling errors. Gaube et al.[27] have used multi-agent methods to decide on land-use changes in four scenarios in the Reichraming region of Austria, aiming to achieve the highest ecological and socio-economic effects. Happe et al.[28] have simulated the changes in land use behavior of farmers under the objective of maximizing benefits and their ecological impacts. Bert et al.[29] have applied multi-agent methods to study the impact of diversity in farmers' decisions on regional landscape structures. Therefore, given the advantages of multi-attribute decision-making inherent in swarm intelligence, focusing our research on the application of multi-agent systems in the video fusion field is a priority.

An agent is an autonomous entity that governs its decision-making behavior within an environment. It interacts with the external environment to collect information, processes this information, and feeds its actions back into the environment. This represents the fundamental function and structure of an agent.

A multi-agent system refers to a distributed intelligent system composed of multiple simple agents that leverage relevant technologies to collaboratively achieve global or local objectives. It not only retains the unique structure and capabilities of individual agents but also demonstrates enhanced intelligence, autonomy, and proactivity. Each agent can collaborate and coordinate with others using appropriate strategies, solving problems in parallel to achieve the overall objective. Moreover, because multi-agent systems are designed in a distributed

manner, diverse, multi-layered agents can be constructed. Their high cohesion and low coupling characteristics provide excellent system scalability. Based on this, we propose an infrared and visible light video fusion method that capitalizes on the features of multi-agent systems, as detailed in Section 3.

## 3. Method

In this section, we first develop a multi-agent video fusion model *MF*. Then, we discuss the perception function and strategy space of the fusion model. Finally, we explore the process of selecting the optimal fusion strategy. The infrared and visible light video fusion algorithm we propose, based on the characteristics of multi-agent systems, analyzes the relationship between multi-agent characteristics and video fusion. According to the fusion requirements, a flexible and variable fusion model is constructed, allowing the fusion strategy to change according to difference feature variations. The multi-agent video fusion model invokes different fusion agents by detecting the main difference features in the video sequences to be fused. The fusion agent calculates the main differential features of the current video frame using the perception function and selects the optimal fusion strategy through the strategy space and utility function. Finally, the fusion strategy is optimized through the interaction and collaboration among different agents to achieve global fusion optimization, as show in Eq (1).

$$MF = \left\{ AF, P, S, U, C, K \right\} \tag{1}$$

Where. *AF* denotes Agent Function, referring to the function set of the agent, encompassing the specific tasks that each agent performs during the fusion process. *P* represents Perception Function, which is used to extract the difference features in video frames and identify the main difference feature of the current frame. *S* represents Strategy Space, which includes the combination of fusion algorithms, fusion rules, and fusion parameters, defining all possible fusion strategies. *U* represents Utility Function, which is used to evaluate the effectiveness of different fusion strategies and select the optimal strategy. *C* represents Collaboration Mechanism, which defines how multiple agents collaborate to optimize the global fusion task. *K* represents the knowledge base, which is used to store the optimal fusion strategies accumulated by the agent during the fusion process for use in future similar tasks. The main process is illustrated in Fig 1.

### 3.1 Perception function of the fusion model

Considering the significant differences in the imaging mechanisms of infrared and visible light sensors and their image characteristics[30], we select six categories of features to measure the differences between the two kinds of videos: Gray Mean (GM), Edge Intensity (EI), Standard Deviation (SD), Average Gradient (AG), Coarseness (CA) and Contrast (CN).

The set of differential feature types is defined as follows according to Eq (2):

$$M = \left\{ \mathrm{m} | GM\, EI\, SD\, AG\, CA\, CN \right\} \tag{2}$$

Difference features represent the absolute differences in the features within corresponding frames of infrared and visible image, as indicated by Eq (3).

$$D_{i,m} = \left| D_{i,m}^{I} - D_{i,m}^{V} \right| \tag{3}$$

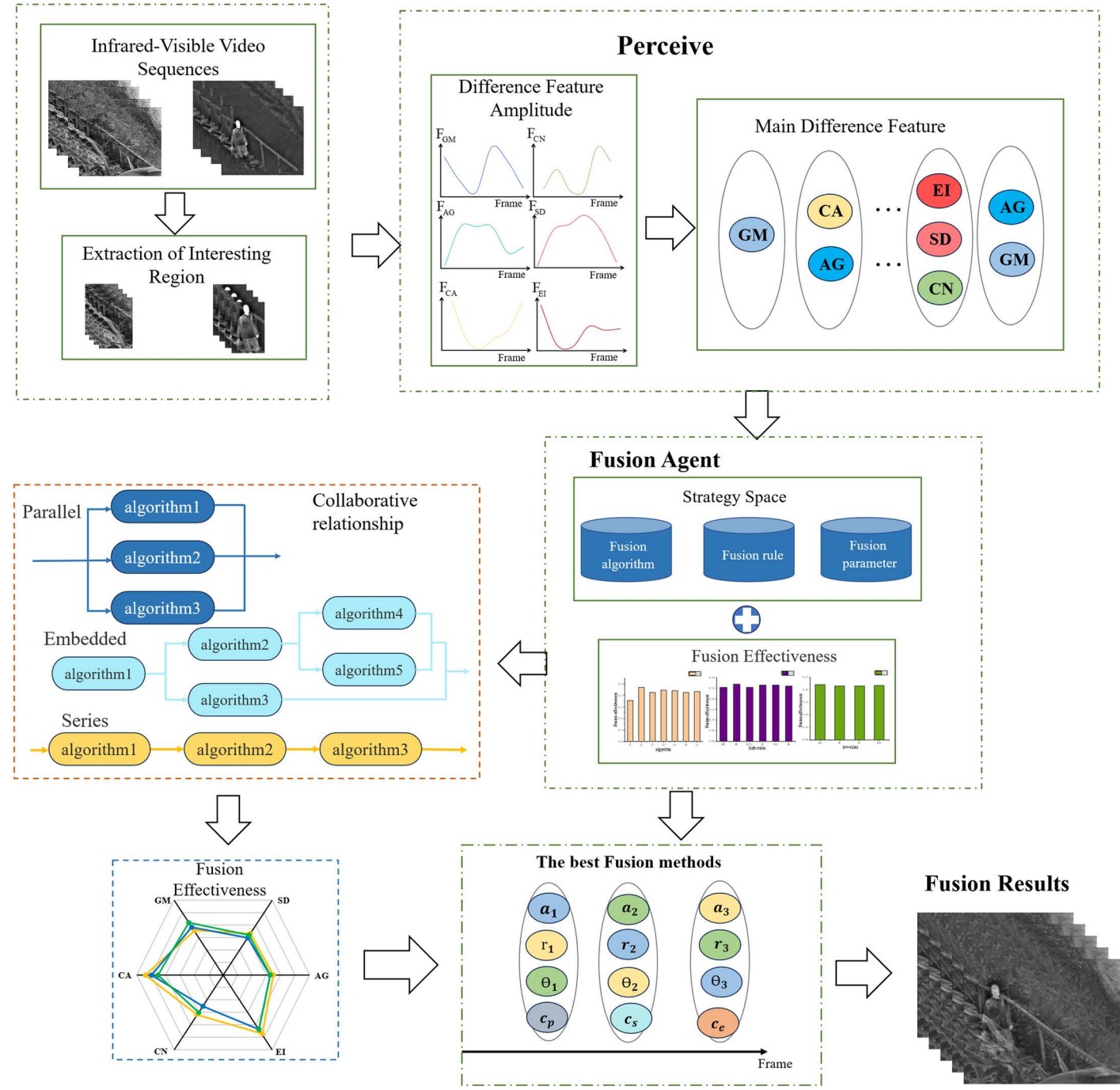

**Fig 1. Overall flowchart of Multi-agent fusion model.**

Where, $D_{i,m}^I$ and $D_{i,m}^V$ respectively denote the magnitude of feature $m$ in the ith frame of the infrared and visible video sequences. $D_{i,m}$ respectively denote the magnitude of the difference feature.

During the operation of the multi-agent video fusion model, the fusion agents first perceive the environment by sensing several types of difference features in the video frames to be

fused, thus determining the main difference feature of the frames. Because the six categories of difference features often vary in dimension, which affects the analysis and comparison of the experimental results. We normalize the difference features of each frame, which makes the comparability between the features greatly enhanced, and it is easier to select the main difference features of the video sequence, the normalized data takes the value range of [0,1]. The feature type corresponding to the maximum normalized difference feature value of the video frame to be fused is regarded as the main difference feature of that frame. We consider this process as the result of the perception function $P(i)$.

The perception function is defined as show in Eq(4):

$$P(i) = \arg\max_{m \in M} \frac{D_{i,m} - min(D_{1,m}, D_{2,m}, \ldots, D_{n,m})}{max(D_{1,m}, D_{2,m}, \ldots, D_{n,m}) - min(D_{1,m}, D_{2,m}, \ldots, D_{n,m})} \tag{4}$$

The multi-agent fusion model utilizes the perception function $P(i)$ to identify the main difference feature type from the input video sequence at frame $i$. Specifically, $P(i)$ outputs the main difference feature type by analyzing the difference characteristics between the infrared and visible light frames. This output $P(i)$ serves as the basis for the fusion agent to subsequently select the optimal fusion strategy, based on the identified primary characteristic.

## 3.2. Construction and evaluation of the strategy space

In the multi-agent fusion model, the strategy space $S$ acts as a fusion method database from which the fusion agent selects the optimal fusion algorithm, rules, and parameters, making its construction essential. To optimize detail retention, contrast enhancement, and the overall visual effect in the image fusion process, it is important to comprehensively select fusion algorithms with varying directionality, sparse representation capabilities, and adaptability. Algorithms like CVT, DTCWT, and NSCT effectively decompose and process image details, edges, and low-frequency information, enhancing the clarity and layering of the fused image. NSST and GFF can smoothly process different image regions, ensuring that the fused image retains its details without losing critical information. Fusion rules and parameters are key to effectively combining different features, ensuring a balance and enhancement of detail, contrast, and overall brightness in the fused image. This multi-dimensional selection of fusion strategies allows the multi-agent fusion model to improve video fusion performance in complex scenes. We selected seven fusion algorithms in the strategy space and optimized the design based on the fusion rules for high-frequency and low-frequency features, along with the key fusion parameters of various algorithms.

The complete strategy combination is shown in Eq(5):

$$S = \left\{ (a, b, \theta) a \in A, r \in R, \theta \in \Theta \right\} \tag{5}$$

The fusion algorithms, fusion rules, and fusion parameters included in the strategy space for our experiment are as follows:

Fusion algorithm $a \in A$: We select seven algorithms in the multi-scale fusion framework, including Curvelet transform (CVT)[31], Dual Tree Complex Wavelet Transform (DTCWT)[32], Laplace Pyramid (LP)[33], Non-Subsampling Contourlet transform(NSCT) [34], Non-Subsampling Shearlet transform (NSST)[35], Guided Filter (GFF)[36] and Wavelet Packet Transform (WPT)[37].

Fusion rule $r \in R$: We chose the high-frequency fusion rule and the low-frequency fusion rule under the multi-scale fusion framework[38, 39], as shown in Table 1.

Fusion parameters $\theta \in \Theta$: The fusion parameters in this paper mainly refers to the number of layers in the multi-scale decomposition of the algorithm and the types of

internal filters. Examples include a layered parameter 'n', where 'n' can take on numeric values '1-7' in LP, DTCWT, and filters such as 'near_sym_a', 'antonini', 'legall'; in the NSST algorithm, shear wave directional parameters (dcomp) and scale parameters (dsize), along with pyramid transformations like 'pyrexc', 'pyr', 'maxflat', '7-9', and others.

### 3.3. Decision-making process of the fusion agent

After perceiving the main difference feature of the video frame to be fused, the multi-agent fusion model activates the fusion agents to select the optimal fusion strategy.

In this process, the fusion agents perform two primary functions: first, they develop and execute local plans to achieve their specific objectives; second, they collaborate with other agents to meet system-level goals[40]. The internal structure of agents should be designed to align with these functions, starting from the tasks they perform. For fusion requirements, encapsulating or mapping logical functions into agents determines the granularity of the agents, and the granularity size impacts the execution efficiency of the entire multi-agent fusion model. In this paper, we design six fusion agents based on the categories of difference features, forming an agent set *FA*, which integrates characteristics of reactive, deliberative, and self-learning agents. These agents are named according to the types of difference features (as shown in Eq(6)).

$$FA = \left\{ A_1, A_2, A_3, A_4, A_5, A_6 \right\} \tag{6}$$

Where $A_1 - A_6$ correspond to different main difference feature fusion agent, each agent is responsible for selecting the optimal fusion strategy for video frames associated with the corresponding main difference feature.

During the selection process, it is essential to evaluate and compare the fusion effects of different algorithms using a unified evaluation standard. Typically, a higher difference feature value indicates better image quality, but comparing these values directly is challenging. To facilitate the observation of fusion effects, we designed utility function to assess the effectiveness of feature fusion between the fused image and the two original images.

Here, the utility function represents the fusion effect on the i-th frame, for the main difference feature m, using fusion strategy S, the larger the value, the higher the fusion effectiveness. as shown in Eq(7).

$$U_{i,m}(a,r,\theta) = w_{i,m}^{\mathrm{I}} \times SIM\left(D_{i,m}, D_{i,m}^{\mathrm{I}}\right) + w_{i,m}^{\mathrm{V}} \times SIM\left(D_{i,m}, D_{i,m}^{\mathrm{V}}\right) \tag{7}$$

**Table 1. Fusion rule set.**

| Fusion rules | High-frequency | Low-frequency |
|---|---|---|
| 1 | Maximum absolute value (MAX) | The maximum of the coefficient (MC) |
| 2 | The maximum of the coefficient (MC) | Weighted mean (SWA) |
| 3 | Frequency selective weighted median filter (FSWM) | Window based standard deviation (WBSD) |
| 4 | Based on the window energy (WE) | Window based weighted average (WBWA) |
| 5 | Principal component analysis (PCA) | |
| 6 | Based on the window gradient (WBG) | |

Where.

$$
\begin{cases}
SIM\left(D_{i,m},D_{i,m}^{\mathrm{I}}\right)=\dfrac{D_{i,m},D_{i,m}^{\mathrm{I}}}{\sqrt{\left(D_{i,m}\right)^2+\left(D_{i,m}^{\mathrm{I}}\right)^2}} \\[3ex]
SIM\left(D_{i,m},D_{i,m}^{\mathrm{V}}\right)=\dfrac{D_{i,m},D_{i,m}^{\mathrm{V}}}{\sqrt{\left(D_{i,m}^{k}\right)^2+\left(D_{i,m}^{\mathrm{V}}\right)^2}}
\end{cases}
\tag{8}
$$

$$
\begin{cases}
\langle D_{i,m},D_{i,m}^{\mathrm{I}}\rangle=D_{i,m}\cdot D_{i,m}^{\mathrm{I}} \\
\langle D_{i,m},D_{i,m}^{\mathrm{V}}\rangle=D_{i,m}\cdot D_{i,m}^{\mathrm{V}}
\end{cases}
\tag{9}
$$

$$
\begin{cases}
w_{i,m}^{I}=\dfrac{D_{i,m}^{\mathrm{I}}}{D_{i,m}^{\mathrm{I}}+D_{i,m}^{\mathrm{V}}} \\[3ex]
w_{i,m}^{V}=\dfrac{D_{i,m}^{\mathrm{V}}}{D_{i,m}^{\mathrm{I}}+D_{i,m}^{\mathrm{V}}}
\end{cases}
\tag{10}
$$

$w_{i,m}^{I}$ and $w_{i,m}^{V}$ denote the respective weights of differential features attributed to the infrared and visible images. $D_{i,m}$ represents the magnitude of the primary differential feature in the fusion result of the i-th frame.

By incorporating the utility function, the fusion agent employs a step-by-step decision-making approach to quantitatively evaluate different candidate strategies and ultimately selects the strategy with the highest utility value, determining the optimal fusion algorithm, rules, and parameters.

First, the fusion agent $A_i$ initializes the fusion rules $r_1$ and fusion parameters $\theta$ for six fusion algorithms from the strategy space. The agent evaluates the performance of each fusion algorithm using the utility function and selects the fusion algorithm a* that maximizes the utility value. Then, based on the selected optimal fusion algorithm a*, the fusion parameters $\theta$ are fixed, and the utility function is used to evaluate each fusion rule, selecting the fusion rule r* that maximizes the utility value. Finally, the agent traverses the parameter set $\Theta$ and calculates the utility value for each fusion parameter. By comparing the effect of each fusion parameter on the fusion result, the optimal parameter $\theta$* is selected, resulting in the optimal fusion strategy, as illustrated in the Eq(11).

$$
S^{\star}=\left\{a^{*},r^{*},\theta^{*}\right\}=\arg\max_{s\in S}U_{i,m}(a,r,\theta)
\tag{11}
$$

In video fusion tasks under complex conditions, fusion agents need to collaborate with each other. Collaboration allows multiple agents to share information and work together, facilitating inter-algorithm fusion. The collaboration methods in the multi-agent fusion model are categorized into three types: serial collaboration, parallel collaboration, and embedded collaboration. These methods exhibit distinct characteristics during agent task execution, as detailed below:

Serial collaboration involves the sequential execution of tasks by fusion agents, where the output of one agent serves as the input for the next. Parallel collaboration refers to different fusion agents performing image fusion simultaneously, with the multiple fusion results being combined to produce the final fusion result, as shown in the equation. Embedded collaboration entails different fusion agents obtaining the final fusion result through a nested structure, where internal outputs can serve as inputs for other fusion algorithms. Once internal fusion is complete, external fusion algorithms integrate the internal fusion results to form the final embedded structure fusion result.

The three types of collaboration relationships mentioned above can be represented as shown in Eq (12).

$$C \in \left\{ C_s, C_p, C_e \right\} \tag{12}$$

In the subsequent fusion process, the similarity between video frames with the same main difference feature can be used to determine whether to directly call the self-learning fusion strategy knowledge base or continue inferring a new fusion strategy. This eliminates the need to infer frame by frame in the video sequence, greatly improving the efficiency of video fusion.

The critical factor in selecting the optimal fusion strategy is the perception function's ability to determine the number of main difference feature types present in the video frames to be fused. If the video frame contains only one main difference feature, a single fusion agent is sufficient to make step-by-step decisions on the fusion algorithm, rules, and parameters, selecting the fusion strategy based on the utility function values. However, when the video frame contains multiple main difference features, collaboration between fusion agents must be considered to achieve global optimization of the fusion effect.

## 4. Experiment and discussions

We will delineate the computational process of our proposed method in this section, outlining the workflow as illustrated in Fig 1. Subsequent experiments are conducted on BEPMS dataset[41], OTCBVS dataset[42] and INO Image Fusion dataset[43], to demonstrate the advantages and robustness of the proposed method. Finally, qualitative and quantitative analyses are used to compare our method with other methods.

### 4.1. Source video datasets

The BEPMS dataset is commonly used for qualitative and quantitative comparisons in the field of image fusion. This dataset involves a scene of a soldier in camouflage walking through dense forest. In this experiment, our focus is on the first 200 frames of this dataset. OTCBVS contains 6 subcategories of datasets, including 17,089 infrared and visible light images from different scenes. In this paper, we use the second subcategory, "Ohio State University (OSU) Color-Thermal Database," with "Location 2" as the experimental group. The scene is located at a busy intersection on the Ohio State University campus, with 6 optical/thermal imaging sequences (3 per location), and the image size is $320 \times 240$ pixels. The INO dataset includes infrared and visible light videos captured under different weather conditions, featuring pedestrians, cars, and lawns in modern urban scenes, which can fully test the target detection capability of the algorithm.

### 4.2. Case study

The method is validated using selected infrared and visible video sequences from the BEPMS dataset.

First, areas of interest of the video sequence are roughly segmented. Here, we select frames 1, 65, 120, 140, 185 and 200 as significant frames for display through manual selection and data analysis, as shown in the red boxes in Fig 2. Secondly, the multi-agent fusion model uses Eq (4) to perceive the video frames to be fused, determining the corresponding main difference feature. Thereby obtaining the main difference feature for video sequences. Finally, based on the perception results of the fusion model, the corresponding fusion agent $A_i$ is activated, and the optimal fusion strategy is selected by combining the strategy space $S$ with the utility

function *U*. For video frames containing multiple primary differential features, achieving global optimization of the fusion process necessitates the collaboration and integration of multiple fusion agents. The main difference feature corresponding to each frame is discernible from the bold numbers presented in Table 2.

The fusion agent first performs a traversal of the fusion algorithms in the strategy space to decide on the optimal fusion algorithm. According to utility function $U_{i,m}(a,r,\theta)$, the distribution of fusion effectiveness for different fusion algorithms corresponding to the main difference features of each significant frame obtained (shown in Fig 3).

The main difference features for Frame 1 include SD, AG and EI, with the optimal algorithms being NSCT, WPT, and DTCWT, respectively (Fig 4 only shows the fusion algorithm for the main difference feature SD of the first frame). For Frame 185, the main difference feature is CA, with the optimal algorithm being DTCWT. Similarly, the optimal fusion algorithms corresponding to the main difference features of the other significant frames can be derived.

After selecting the optimal fusion algorithm, the fusion agent determines the optimal fusion rules for the main difference features of the images to be fused, in conjunction with fusion effectiveness. First, the low-frequency fusion rules and parameters are set, and then all combinations of high-frequency fusion rules are fused. Subsequently, with the fusion parameters and previously determined high-frequency fusion rules set, all combinations of low-frequency fusion rules are fused. The results of the fusion are depicted in Fig 4 and Fig 5.

Finally, with fusion parameters as variables, the fusion agent sequentially compares the fusion effects of each parameter based on fusion effectiveness. The analysis process is as above.

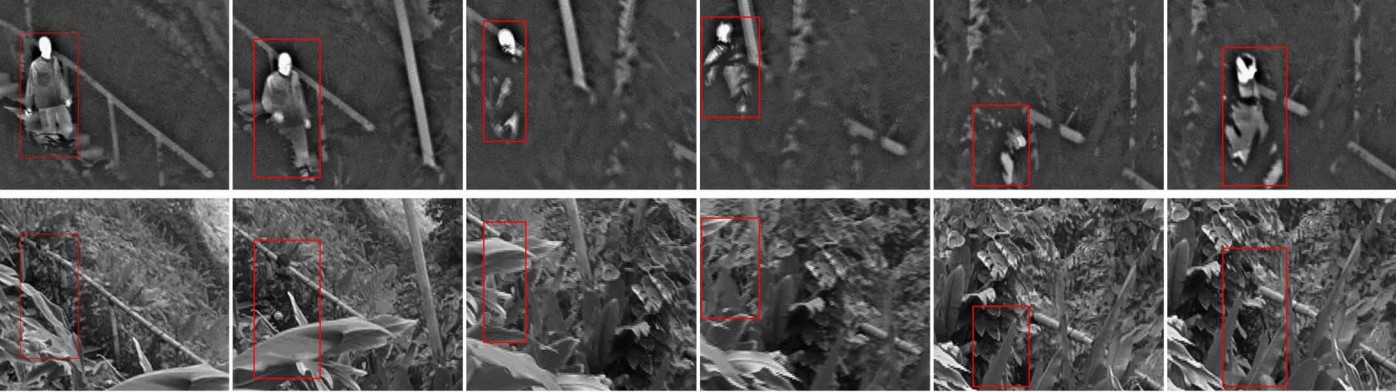

**Fig 2. Significant frames of infrared and visible videos in the BEPMS dataset.** From left to right they are: frame 1, frame 12, frame 50, frame 105, frame 185 and frame 200.

**Table 2. Normalized results of significant frame difference features.**

| Difference feature | Infrared-visible Video Sequences | | | | | |
|---|---|---|---|---|---|---|
| | 1 | 65 | 120 | 140 | 185 | 200 |
| GM | 0.3176 | **0.5947** | **1.0000** | 0.0490 | 0.4728 | 0.0967 |
| SD | **1.0000** | 0.2643 | 0.3096 | 0.4923 | 0.2907 | 0.4437 |
| AG | **1.0000** | 0.2387 | 0.1151 | 0.0000 | 0.2156 | 0.5014 |
| EI | **1.0000** | 0.4826 | 0.3067 | 0.0000 | 0.2832 | **0.5913** |
| CN | 0.3957 | 0.4493 | 0.4568 | **1.0000** | 0.4729 | 0.4481 |
| CA | 0.6241 | 0.4569 | 0.9767 | 0.6842 | **1.0000** | 0.1370 |

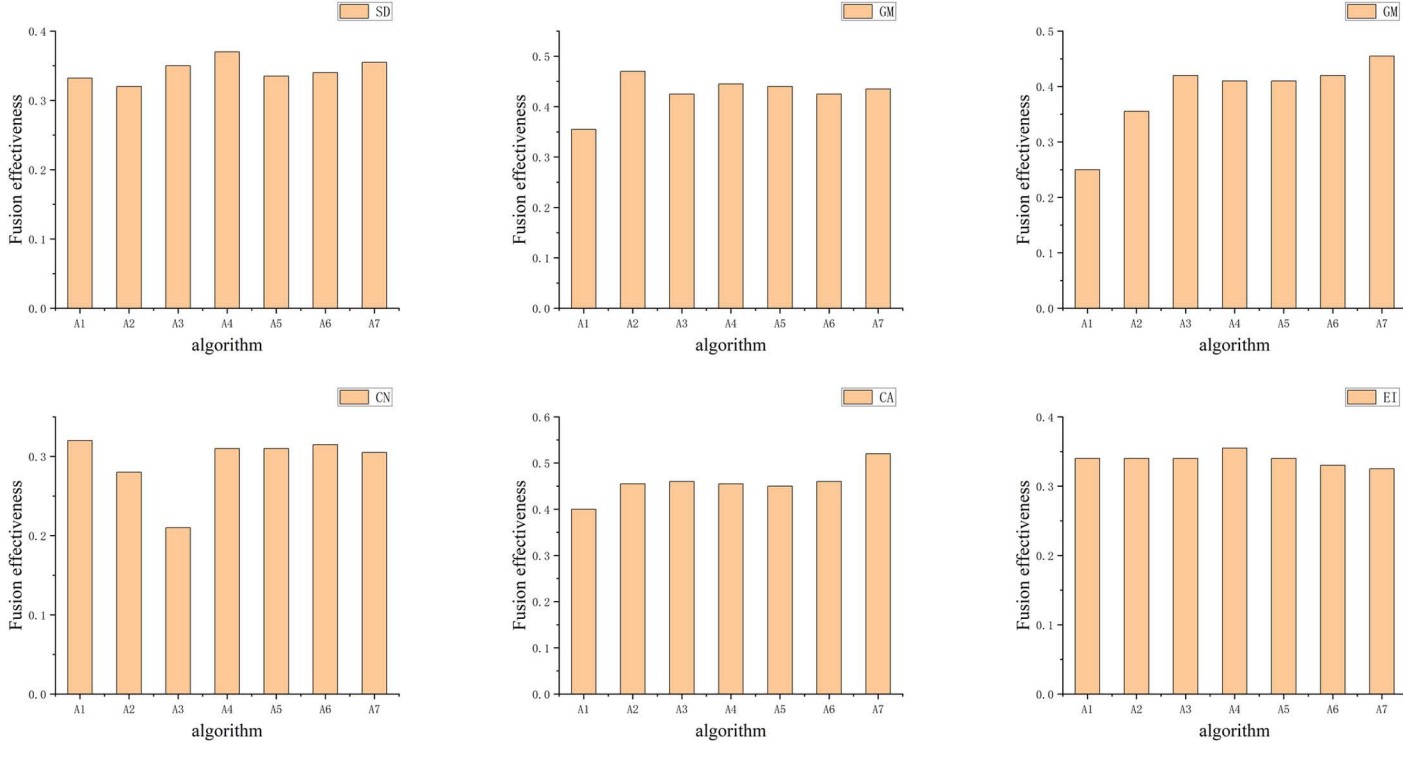

**Fig 3. Fusion effectiveness of 6 groups of images.**

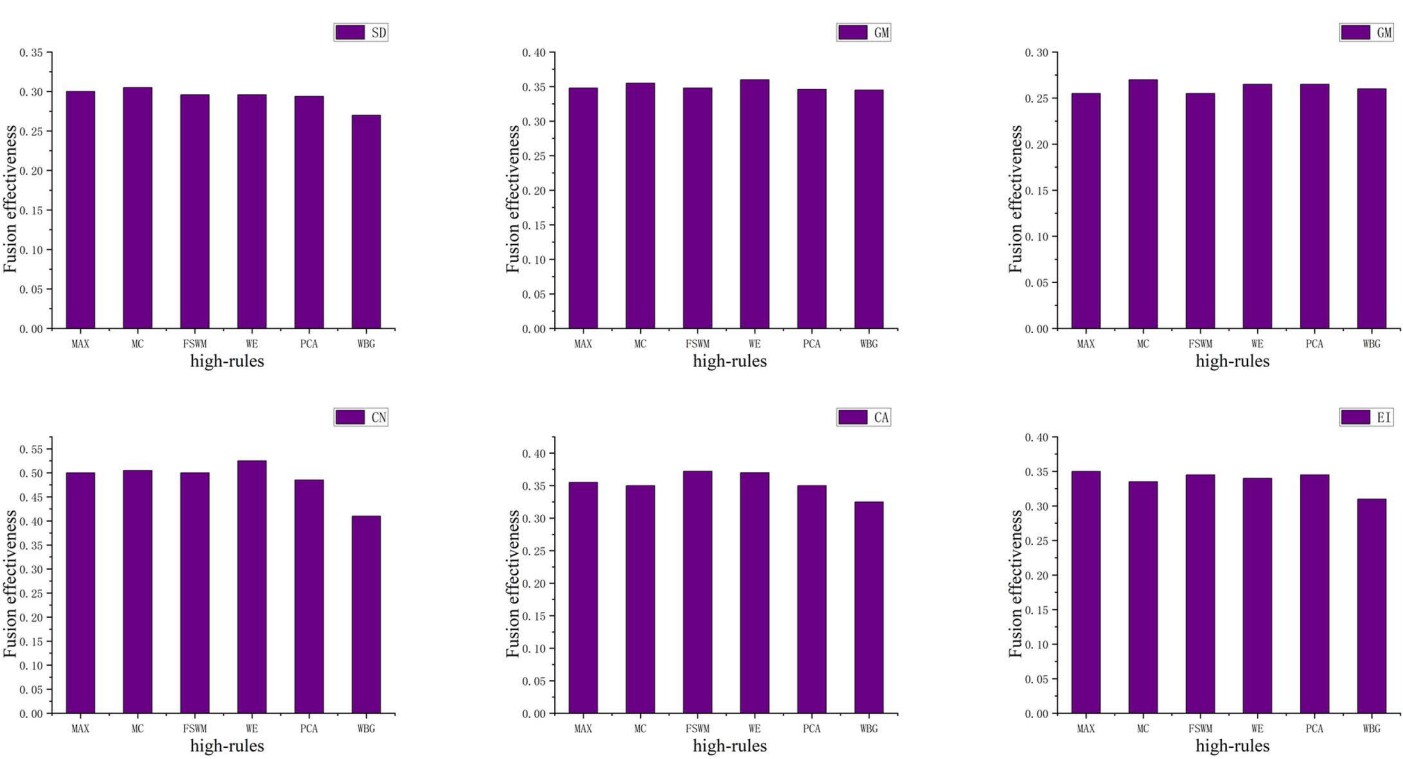

**Fig 4. Effectiveness of fusion in High-frequency fusion rules.**

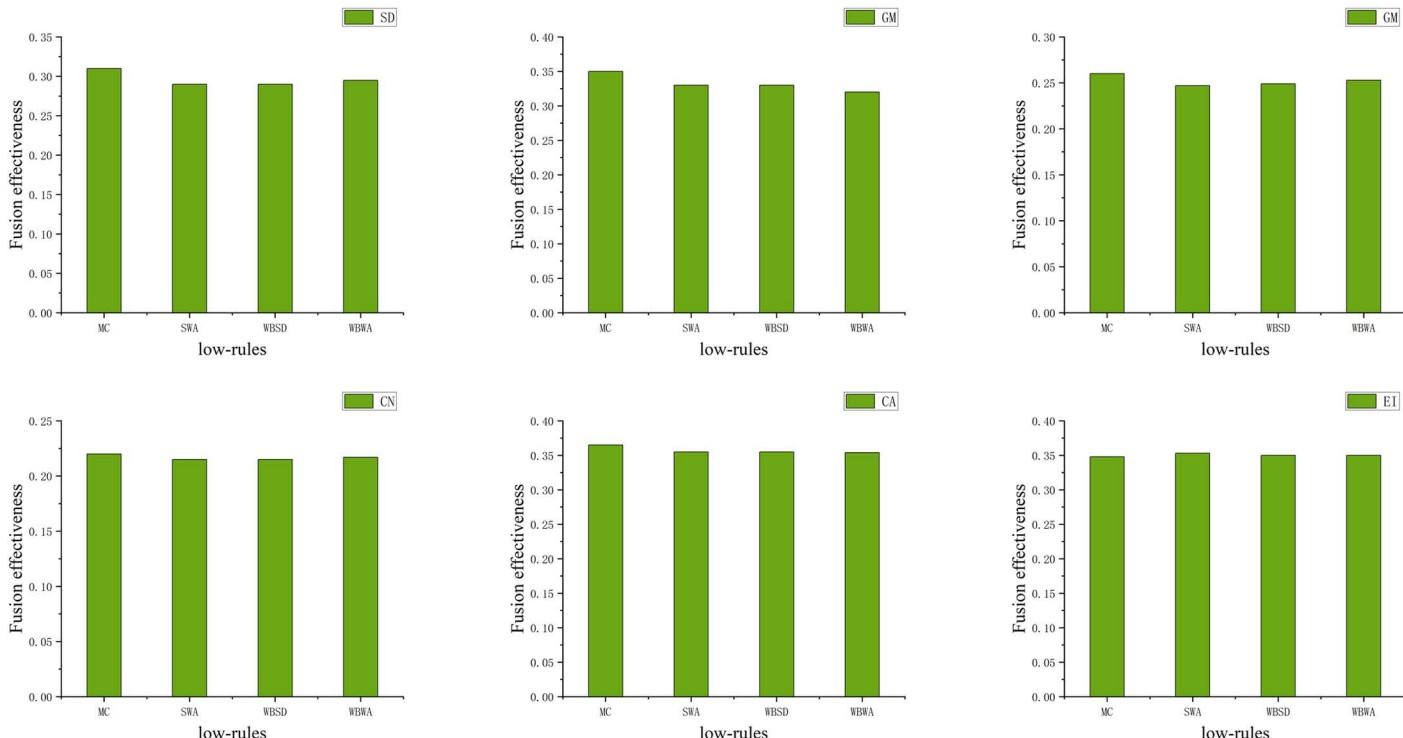

**Fig 5. Effectiveness of fusion in Low-frequency fusion rules.**

In conclusion, the results for the optimal fusion strategies are shown in Table 3.

When multiple main difference features are present within a single frame, the optimal fusion strategy chosen by the corresponding fusion agents differs, making the process more complex. At this point, a single fusion agent cannot complete the image fusion process alone. It requires leveraging the collaborative interactions among multiple fusion agents to combine different fusion strategies effectively. By comparing the fusion effectiveness of several types of collaborative relationships, the optimal fusion structure is determined. For instance, in the first frame of the above experiment, the main difference features are SD, AG and EI; consider the fusion effectiveness of these three fusion agents under different collaborative methods.

The experiment is as follows: Series: First, the SD fusion agent is used to fuse the images to be fused. Next, the EI fusion agent perceives the fusion result of the SD fusion agent and the infrared video frame to continue to obtain the fusion result. Finally, the AG fusion agent perceives the above fusion result and the visible video frame to obtain the final fusion result. Parallel: The fusion results of the three fusion agents are averaged and weighted to obtain the final fusion result. Embedded: The SD fusion agent is used to decompose the images to be fused to obtain high-frequency image and low-frequency image. The low-frequency image is decomposed by the EI fusion agent, and the high-frequency image is decomposed by the AG fusion agent. Finally, the final fusion result is obtained by layer-by-layer reconstruction.

For Frame 1, the fusion effectiveness under the serial, parallel and embedded collaborative methods is calculated, and serial is determined to be the optimal method of collaboration, as shown in Fig 6.

**Table 3. The optimal fusion strategies corresponding to each frame.**

| | Infrared-visible Video Sequences | | | | | |
|---|---|---|---|---|---|---|
| | 1 | 65 | 120 | 140 | 185 | 200 |
| **Algorithm** | NSCT/WPT/DTCWT | DTCWT | WPT | CVT | DTCWT | NSCT |
| **Rule** | MC-WA/WBG-WA/WE-WBWA | WBG-WA | MC-WA | WBG-WA | WE-WA | MAX-MC |
| **Parameter** | '9-7'/n = 3/near_sym_a | antonini | n = 7 | n = 2 | 'near_sym_a | '7-9' |

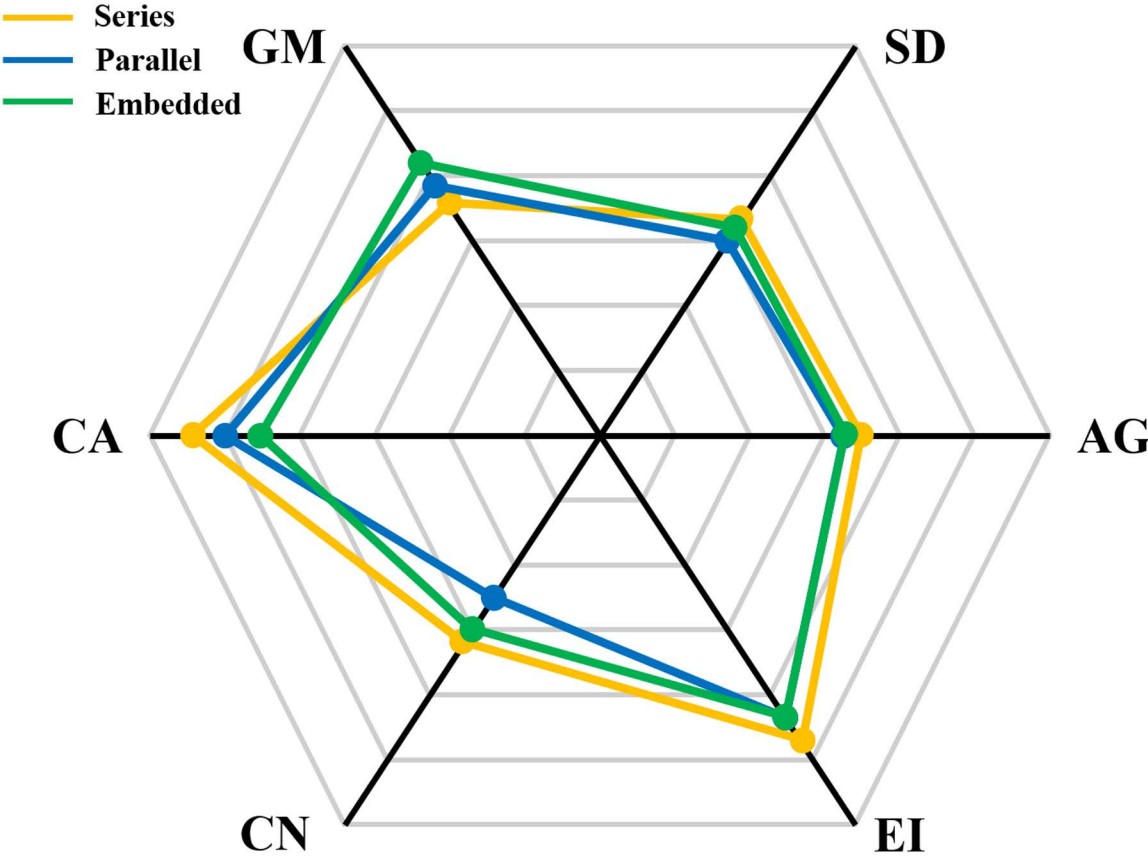

**Fig 6. Fusion performance of 3 groups fusion structures.**

## 4.3. Discussion of experimental results

In order to verify the rationality and effectiveness of the proposed algorithm, we validate the fusion performance through both qualitative and quantitative analyses. The fusion results of the proposed method are compared with the multi-scale transform algorithms selected in Section 3 (CVT, DTCWT, LP, NSCT, NSST, GFF, and WPT) and the NestFuse[44] fusion model based on neural networks. All methods follow the parameter settings specified in the original papers. All experiments in this paper were conducted on a Windows 10 system with an Intel (R) Core (TM) i5-8300U, and all algorithms were implemented in MATLAB 2021a.

The fusion results of each frame, based on the selected optimal fusion strategy, are discussed. Subjective evaluations are prone to influence from the evaluator's personal psychology

and mental state, introducing a degree of subjectivity. Therefore, we selected 7 objective evaluation metrics to evaluate the fusion results of each method, and the fusion effect is directly proportional to the metric values. In the experiments, the best and second-best values are highlighted in bold black italics and bold black, respectively.

**4.3.1. Metric.** The evaluation metrics include: the fusion volume function ($Q^{AB/F}$)[45], the weighted quality assessment metric ($Q_w$)[46], the universal image quality index ($Q_0$)[47], a structure similarity-based metric ($Q_e$)[48], mutual information ($MI$)[49], spatial frequency ($SF$)[50], and visual information fidelity ($VIFF$)[51].

$Q^{AB/F}$ is a function that represents the information from each source image in the fused image. The higher the $Q^{AB/F}$ value, the more information from each source image is retained. $Q^{AB/F}$ ranges from 0 to 1, with $Q^{AB/F}$ = 0 indicating total loss of input information and $Q^{AB/F}$ = 1 representing an ideal fusion result with no loss of input information. $Q_0$ measures the structural distortion between the fused image and the source images; the closer the value is to 1, the better the fusion performance. The $Q_w$ metric reflects the degree to which the fused image retains information from the original images, as well as their combined effect. The value ranges from 0 to 1, with higher values indicating better image quality and fuller retention of source image information. $Q_e$ is used to evaluate the edge-based fusion performance, with values ranging from 0 to 1. Higher similarity is indicated by values closer to 1. $MI$ measures the fusion results of infrared and visible light images, with the average value taken as the final output. With higher values indicating that more source image features are included in the fusion result. $SF$ is based on gradient distribution and effectively reveals image details and textures. $VIFF$ reflects the distortion in visual information between the fused image and the source images from the perspective of the human visual system. A larger value indicates better fusion quality.4.3.2 Results on BEPMS dataset

The fusion results of the above experiments are qualitatively and quantitatively analyzed for various fusion methods on selected video frames (as show in Fig 7). It is clear that a single algorithm cannot ensure optimal results over the entire video sequence. The performance of the fusion algorithm varies with changes in frame differences or visual content. Our method can optimize the selection of fusion strategies based on changes in the main difference features of the areas of interest, thereby maximizing the retention of thermal target information in infrared images and texture detail information in visible images. Among these traditional and advanced methods, each has its advantages, preserving much of the original image information and some scene details. However, these methods do not effectively maintain a balance

**Table 4. The quantitative experimental results on the BEPMS dataset.**

| Fusion Method | Evaluation Index | | | | | | |
|---|---|---|---|---|---|---|---|
| | $Q^{AB/F}$ | $Q_w$ | $Q_0$ | $Q_e$ | $MI$ | $VIFF$ | $SF$ |
| CVT | 0.2712 | *0.8317* | 0.4056 | 0.0845 | 1.5772 | 0.2813 | **14.4261** |
| DTCWT | 0.3469 | 0.6441 | 0.4624 | 0.1483 | 1.1149 | 0.2725 | 13.6647 |
| LP | 0.3549 | 0.6197 | 0.4723 | 0.1404 | 1.1915 | *0.3226* | 14.2224 |
| NSCT | 0.2875 | 0.5886 | 0.4721 | 0.1293 | 2.1204 | 0.2824 | 13.9504 |
| NSST | 0.3627 | 0.6572 | 0.4674 | **0.2074** | 2.1689 | 0.2831 | 14.3368 |
| SWT | 0.3049 | 0.5955 | 0.4766 | 0.1345 | 2.2593 | 0.2866 | 13.5462 |
| WPT | 0.286 | 0.5945 | 0.4182 | 0.1124 | 2.1705 | 0.2916 | 14.0764 |
| NestFuse(avg) | **0.5147** | 0.8052 | 0.5553 | *0.2406* | *2.8138* | **0.3157** | 14.003 |
| NestFuse(max) | 0.5090 | 0.8021 | **0.5565** | 0.1628 | 2.6875 | 0.3040 | 13.6311 |
| Ours | *0.5709* | **0.8289** | *0.5947* | *0.2406* | **2.7833** | 0.3001 | *14.8736* |

in fusion performance. The quantitative results, displayed in Table 4, show that our method attains the highest values on indices $Q^{AB/F}$, $Q_e$, $Q_0$ and $SF$, ranks second to CVT and Nest-Fuse(avg) on $Q_w$ and $MI$, and third on $VIFF$. These results affirm that our approach not only retains an abundance of valid feature information, leading to superior image contrast and visual effects, but also surpasses other methods in fusion performance. This consistency aligns with our qualitative analysis, validating the efficacy of our proposed fusion method.

**4.3.3. Results on OTCBVS dataset.** From the fusion results of the video sequence (as shown in Fig 8). Compared to other methods, the approach described in this paper effectively preserves pedestrian brightness and scene details, offering a clearer visual effect. Qualitative analysis results demonstrate its advantages in retaining key thermal targets and rich visible details. Table 5 displays the quantitative experimental results for the OTCBVS dataset, where our method achieved the highest scores on evaluation indices $Q^{AB/F}$, $Q_w$, $Q_e$, $VIFF$ and $SF$, and ranked second only to NestFuse (avg) on index $MI$.

**4.3.4. Results on INO image fusion dataset.** In order to verify the robustness and effectiveness of the method proposed in this paper, we conducted further tests of the multi-agent collaborative fusion model on the INO dataset, with various fusion results shown in Fig 9. In terms of local details, the results of this study reveal textures that are more realistic and clearer, such as the texture of moving cars. In summary, the method described in this paper demonstrates outstanding fusion performance. Table 6 provides the quantitative experimental results on the INO dataset, achieving top scores in evaluation indices $Q^{AB/F}$, $Q_w$, $Q_0$, $MI$ and $SF$.

Across multiple datasets, the fusion performance of the method proposed in this paper outperforms other approaches. The improvements in the best metrics are consistent, demonstrating that our method is both robust and effective.

When combining the fusion results from three datasets, although our method does not achieve the maximum values across all evaluation metrics, the optimal values represent 66.7% of the total, while sub-optimal values make up 80.9%. This indicates that our method can adaptively select fusion strategies based on the difference features, enabling effective fusion of source image differences. The fusion results show clear superiority compared to other single fusion algorithms.

## 5. Conclusion

This paper introduces an infrared and visible video fusion algorithm based on multi-agent characteristics, conducting a multi-agent video fusion model. Within this model, single fusion

**Table 5. Results of each evaluation index of the OTCBVS dataset.**

| Fusion Method | Evaluation Index | | | | | | |
|---|---|---|---|---|---|---|---|
| | $Q^{AB/F}$ | $Q_w$ | $Q_0$ | $Q_e$ | $MI$ | $VIFF$ | $SF$ |
| CVT | 0.3029 | 0.5574 | 0.3649 | 0.1301 | 2.5776 | 0.1967 | 17.601 |
| DTCWT | 0.3728 | 0.6074 | 0.4522 | 0.21 | 1.9659 | 0.1531 | 14.539 |
| LP | **0.4847** | 0.6843 | *0.5091* | **0.2607** | 2.3333 | 0.2699 | 20.5016 |
| NSCT | 0.4052 | 0.6489 | **0.5025** | 0.2329 | 2.3628 | 0.2621 | 16.1869 |
| NSST | 0.4620 | **0.7121** | 0.4856 | 0.2605 | 2.2325 | **0.2831** | **21.4778** |
| GF | 0.418 | 0.6527 | 0.5047 | 0.2319 | 2.3532 | 0.2662 | 16.7404 |
| WPT | 0.2748 | 0.5494 | 0.3896 | 0.1574 | 2.1436 | 0.2095 | 14.5702 |
| NestFuse(avg) | 0.4612 | 0.6178 | 0.4687 | 0.2423 | *3.1508* | 0.2127 | 15.967 |
| NestFuse(max) | 0.4523 | 0.6213 | 0.4712 | 0.2384 | 3.0169 | 0.2195 | 15.4944 |
| ours | *0.4945* | *0.7246* | 0.4751 | *0.2684* | 3.0536 | *0.2896* | *22.7807* |

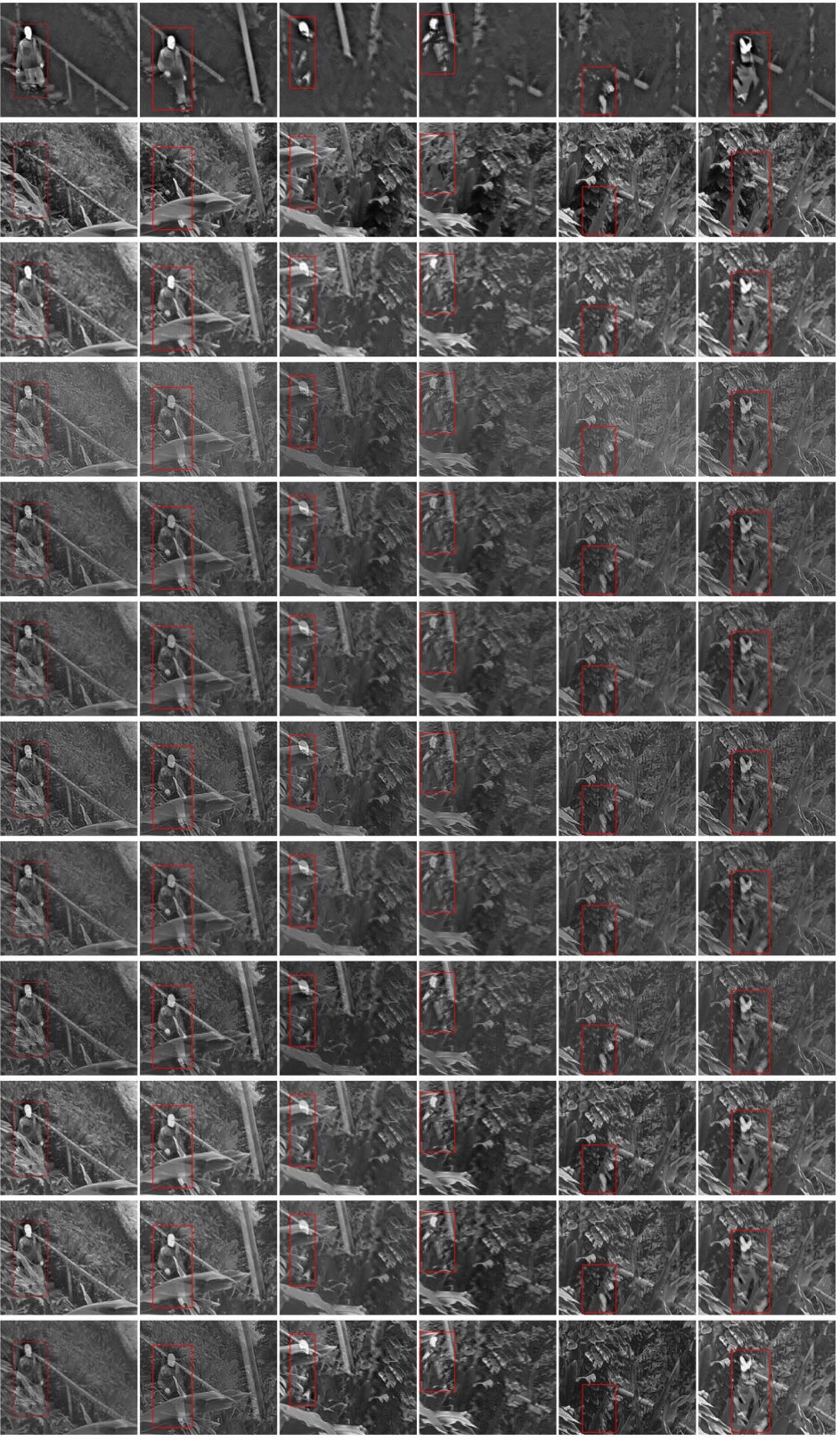

**Fig 7. Fusion outcomes for select frames in the BEPMS dataset.** Each frame sequence from top to down includes: infrared image, visible image, CVT, DTCWT, LP, NSCT, NSST, GF, WPT, NestFuse (avg), NestFuse (max) and results of ours.

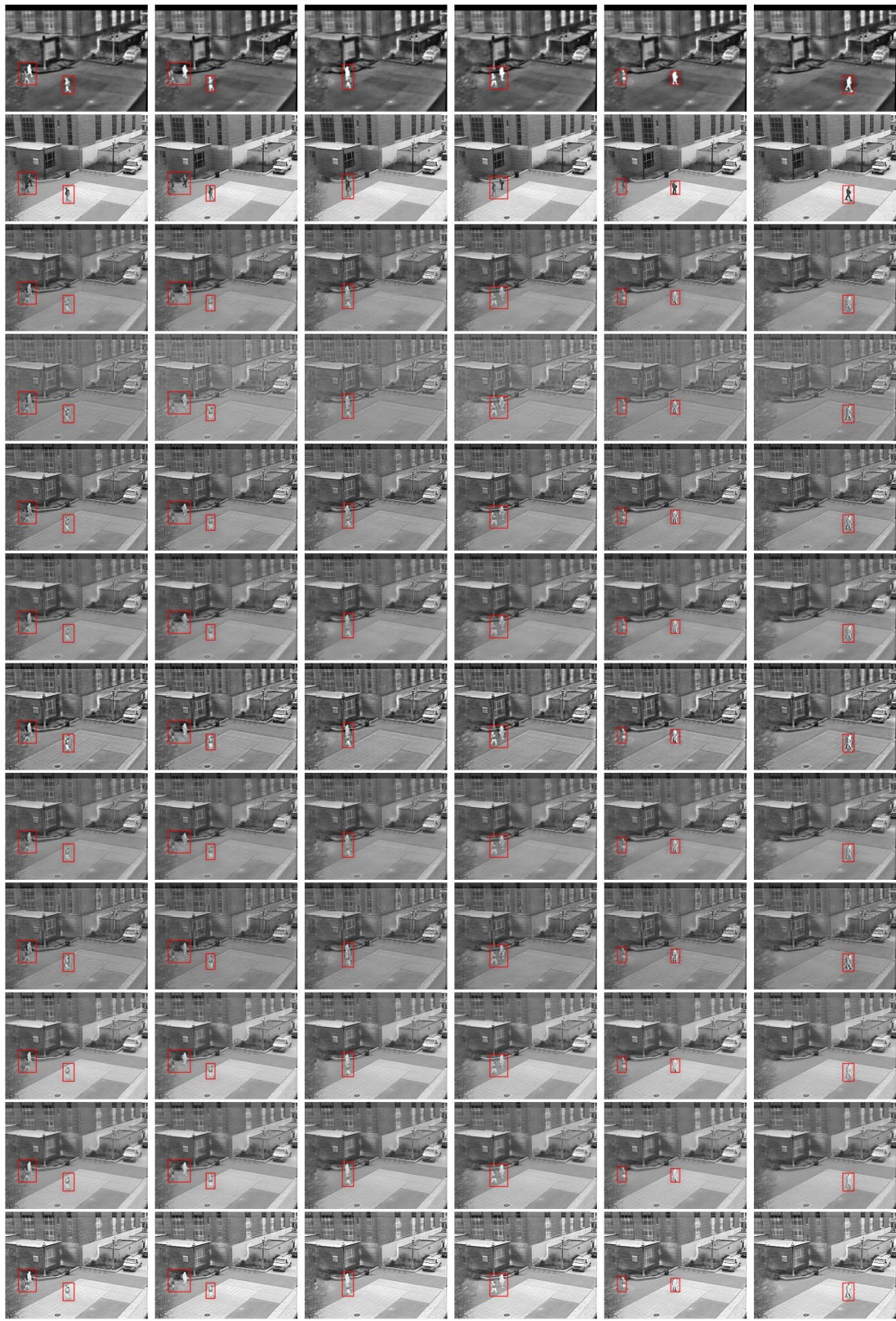

**Fig 8. Fusion outcomes for select frames in the OTCBVS dataset.** Each frame sequence from top to down includes: infrared image, visible image, CVT, DTCWT, LP, NSCT, NSST, GF, WPT, NestFuse (avg), NestFuse (max) and results of ours.

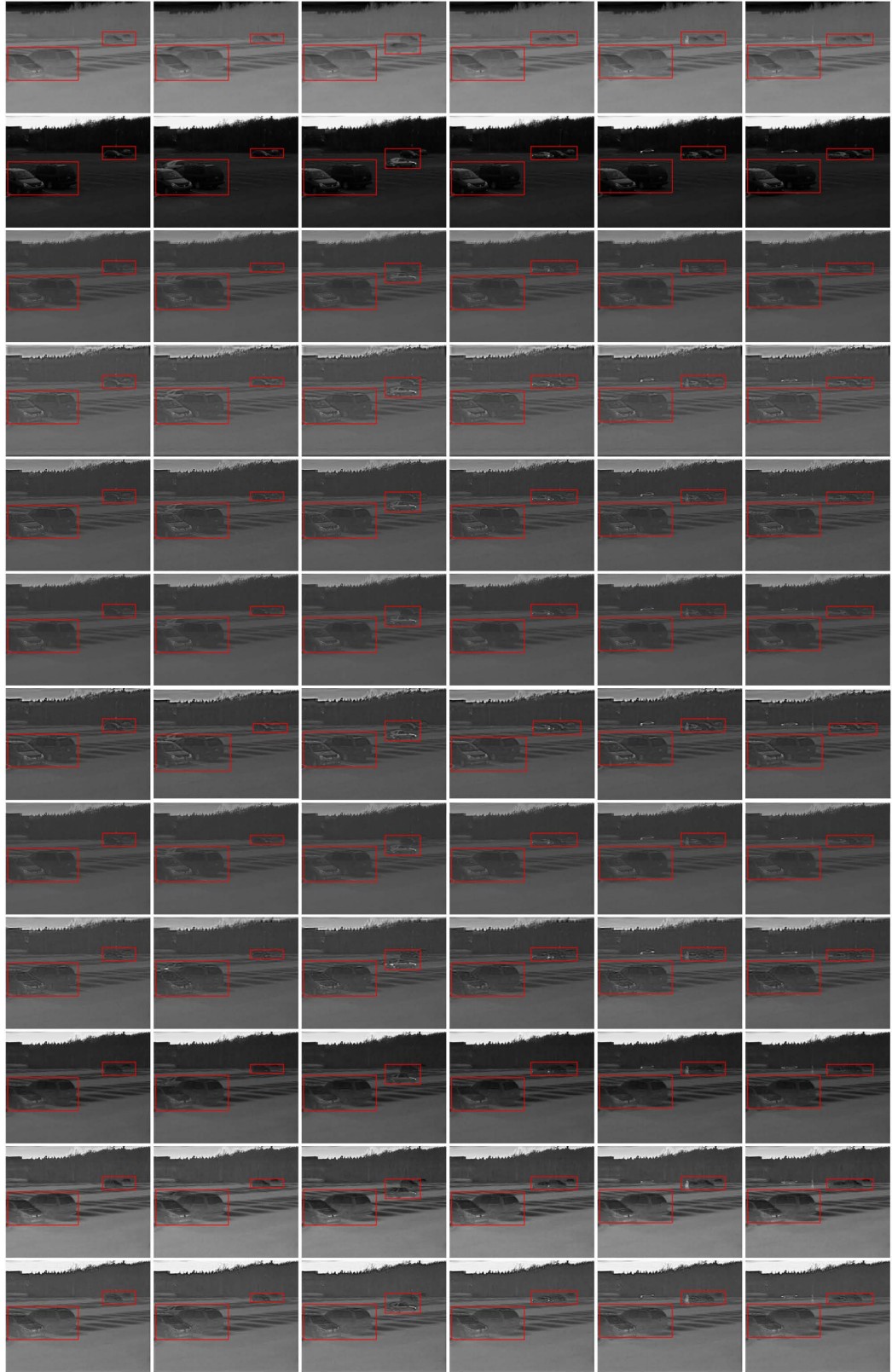

**Fig 9. Fusion outcomes for select frames in the INO dataset.** Each frame sequence from top to down includes: infrared image, visible image, CVT, DTCWT, LP, NSCT, NSST, GF, WPT, NestFuse (avg), NestFuse (max) and results of ours.

Table 6. Results of each evaluation index of the INO dataset.

| Fusion Method | Evaluation Index | | | | | | |
|---|---|---|---|---|---|---|---|
| | $Q^{AB/F}$ | $Q_w$ | $Q_0$ | $Q_e$ | MI | VIFF | SF |
| CVT | 0.3345 | 0.6442 | 0.2349 | 0.0814 | 3.2966 | 0.2923 | **7.4437** |
| DTCWT | 0.4803 | 0.7283 | 0.3237 | 0.2297 | 3.2657 | 0.3168 | 6.3147 |
| LP | 0.4832 | 0.7021 | 0.3815 | 0.2581 | 3.6251 | 0.3545 | 5.2862 |
| NSCT | 0.3879 | 0.6171 | 0.3623 | 0.2203 | **3.9557** | 0.3216 | 3.9909 |
| NSST | 0.5036 | 0.7614 | 0.3625 | **0.2879** | 3.4912 | **0.3851** | 6.0461 |
| GF | 0.4010 | 0.6265 | 0.3538 | 0.2101 | 3.6985 | 0.3251 | 4.7265 |
| WPT | 0.4046 | 0.6733 | 0.3348 | 0.1736 | 3.615 | 0.3578 | 5.8859 |
| NestFuse(avg) | **0.5403** | 0.7564 | 0.3752 | *0.3105* | 3.9256 | *0.5332* | 7.4179 |
| NestFuse(max) | 0.5317 | **0.8093** | **0.3947** | 0.2531 | 3.9362 | 0.3754 | 6.0082 |
| ours | *0.5423* | *0.8153* | *0.3933* | 0.2627 | *4.0158* | 0.3618 | *7.5718* |

agents select the optimal fusion strategy for the main difference features based on utility function. For cases with more than one main difference feature, the model selects the optimal fusion structure through collaborative interaction between agents. Fusion model tailored to the combination of changes in fusion algorithms, fusion rules, fusion parameters, and fusion structures, significantly improves the quality of infrared and visible video fusion. It addresses the problem that existing fusion models cannot be dynamically adjusted according to the video frame difference characteristics, resulting in ineffective fusion.

The fusion results from the BEPMS, OTCBVS, and INO datasets demonstrate that the proposed method effectively preserves infrared typical targets and visible structural details throughout the video. It achieves efficient fusion of difference information and adaptability in image fusion, with performance that significantly outperforms other single fusion methods.

## Author contributions

**Conceptualization:** Yandong Liu, Linna Ji.

**Funding acquisition:** Linna Ji.

**Methodology:** Linna Ji, Fengbao Yang.

**Supervision:** Fengbao Yang.

**Validation:** Yandong Liu.

**Writing – original draft:** Yandong Liu.

**Writing – review & editing:** Xiaoming Guo.

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
