## [Decision Letter · Decision Letter 0]

2 Sep 2024

PONE-D-24-28516MAF: An algorithm based on multi-agent characteristics for infrared and visible video fusionPLOS ONE

Dear Dr. Ji,

Thank you for submitting your manuscript to PLOS ONE. After careful consideration, we feel that it has merit but does not fully meet PLOS ONE’s publication criteria as it currently stands. Therefore, we invite you to submit a revised version of the manuscript that addresses the points raised during the review process.

We look forward to receiving your revised manuscript.

Kind regards,

Yawen Lu, Ph.D

Academic Editor

PLOS ONE

“This work was supported  by  Basic Research Program of Shanxi Province (202203021221104).”

“This work was supported in part by Basic Research Program of Shanxi Province (202203021221104).”

“This work was supported  by  Basic Research Program of Shanxi Province (202203021221104).”

Additional Editor Comments:

PONE-D-24-28516

The reviewers have provided feedback on the authors' work on MAF: An algorithm based on multi-agent characteristics for infrared and visible video fusion. The strengths of the paper include the theoretical formulation and interesting applications. However, the reviewers raise several concerns that need to be addressed in the revision, including but not limited to:

More details on the practical implementation, component description and algorithm steps, etc. More complete literature reviews on multi-agent and multi-modal fusion works.

Overall, a major revision is recommended to address the mentioned concerns and strengthen the paper.

Reviewers' comments:

Reviewer's Responses to Questions

**Comments to the Author**

1. Is the manuscript technically sound, and do the data support the conclusions?

Reviewer #1: Partly

Reviewer #2: Yes

2. Has the statistical analysis been performed appropriately and rigorously? 

Reviewer #1: Yes

Reviewer #2: Yes

3. Have the authors made all data underlying the findings in their manuscript fully available?

Reviewer #1: Yes

Reviewer #2: Yes

4. Is the manuscript presented in an intelligible fashion and written in standard English?

Reviewer #1: No

Reviewer #2: Yes

5. Review Comments to the Author

Reviewer #1: This paper proposes a multi-agent fusion method for infrared and visible video fusion. The authors aim to address limitations of existing fusion models by enabling dynamic adjustment of fusion strategies based on differences between video frames.

Here are some suggestions:

1. The description of the multi-agent system and fusion agents lacks technical details. The authors should provide a more rigorous mathematical formulation of their proposed model.

2. The structure of the paper should be improved for clarity. The methods section in particular jumps between concepts without a clear logical flow.

3. More details are needed on the datasets used.

4. The paper lacks some related fusion papers in the literature review, includes [Mitigating modality discrepancies for RGB-T semantic segmentation][Superthermal: Matching thermal as visible through thermal feature exploration][Matching as color images: Thermal image local feature detection and description], etc.

Reviewer #2: 1.In the introduction section, elaborate in more detail on the limitations of infrared and visible light video fusion in existing methods, particularly the issue of dynamically adjusting fusion strategies. Clearly indicate the potential and advantages of multi-agent systems in solving this problem. Cite more recent literature on multi-agent systems and video fusion in the introduction to enhance the theoretical foundation of the paper. For example: FusionPID, FusionCPP, FusionJPSI, FusionIPCS, ICIF, etc.

2. Provide a detailed description of the various components and interaction mechanisms of a multi-agent system, including how to define the functions of the agent, how to perceive the environment, and how to make decisions.

3. Provide more detailed algorithm steps, including how to select appropriate fusion algorithms, rules, parameters, and structures based on differential features. Add pseudocode or flowcharts to help understand.

4. Explain why these specific fusion algorithms and parameters were chosen, and discuss the importance of these choices in improving fusion performance. Meanwhile, explain how to evaluate the effectiveness of the model, including the selection of evaluation indicators and calculation methods.

5. Provide a detailed list of all parameter settings, hardware configurations, and software environments used during the experimental process, so that other researchers can reproduce the experimental results.

6. PLOS authors have the option to publish the peer review history of their article (what does this mean? ). If published, this will include your full peer review and any attached files.

**Do you want your identity to be public for this peer review?** For information about this choice, including consent withdrawal, please see our Privacy Policy .

Reviewer #1: No

Reviewer #2: **Yes: ** Linlu Dong

---

## [Author Response · Author response to Decision Letter 1]

20 Oct 2024

Dear Editor and Reviewers：

Thank you very much for your valuable comments and suggestions on our manuscript. We have carefully addressed all the points raised by the reviewers and the editor. The detailed, point-by-point responses have been provided in the attached file, which has been uploaded along with the revised manuscript.

Please refer to the attached document for the full responses and explanations of the revisions.

Thank you again for your time and consideration. We look forward to your feedback.

Yours sincerely,

Linna Ji

---

## [Decision Letter · Decision Letter 1]

17 Nov 2024

PONE-D-24-28516R1MAF：An algorithm based on multi-agent characteristics for infrared and visible video fusionPLOS ONE

Dear Dr. Ji,

Thank you for submitting your manuscript to PLOS ONE. After careful consideration, we feel that it has merit but does not fully meet PLOS ONE’s publication criteria as it currently stands. Therefore, we invite you to submit a revised version of the manuscript that addresses the points raised during the review process.

The reviewers have provided feedback on the authors' work on "MAF：An algorithm based on multi-agent characteristics for infrared and visible video fusion". Please fix the remaining issues on the proofreading and format of the submission.

Overall, a Minor Revision is recommended to address the minor issues in the paper.

We look forward to receiving your revised manuscript.

Kind regards,

Yawen Lu, Ph.D

Academic Editor

PLOS ONE

Journal Requirements:

Additional Editor Comments :

Dear Linna Ji,

PONE-D-24-28516R1

The reviewers have provided feedback on the authors' work on "MAF：An algorithm based on multi-agent characteristics for infrared and visible video fusion". Please fix the remaining issues on the proofreading and format of the submission.

Overall, a Minor Revision is recommended to address the minor issues in the paper.

Reviewers' comments:

Reviewer's Responses to Questions

**Comments to the Author**

1. If the authors have adequately addressed your comments raised in a previous round of review and you feel that this manuscript is now acceptable for publication, you may indicate that here to bypass the “Comments to the Author” section, enter your conflict of interest statement in the “Confidential to Editor” section, and submit your "Accept" recommendation.

Reviewer #1: All comments have been addressed

Reviewer #2: All comments have been addressed

2. Is the manuscript technically sound, and do the data support the conclusions?

Reviewer #1: Yes

Reviewer #2: Yes

3. Has the statistical analysis been performed appropriately and rigorously? 

Reviewer #1: Yes

Reviewer #2: Yes

4. Have the authors made all data underlying the findings in their manuscript fully available?

Reviewer #1: Yes

Reviewer #2: Yes

5. Is the manuscript presented in an intelligible fashion and written in standard English?

Reviewer #1: No

Reviewer #2: Yes

6. Review Comments to the Author

Reviewer #1: Most of the comments have been addressed. Here are some additional suggestions.

-please carefully proofread the manuscript to ensure clarity, consistency, and accuracy.

-Also, kindly double-check the citation format to make sure all references adhere to the required style guidelines.

Reviewer #2: 1.If the authors have adequately addressed your comments raised in a previous round of review and you feel that this manuscript is now acceptable for publication, you may indicate that here to bypass the “Comments to the Author” section, enter your conflict of interest statement in the “Confidential to Editor” section, and submit your "Accept" recommendation. Your response was excellent, it resolved my concerns, and now it can be published.yes

2.The manuscript must describe a technically sound piece of scientific research with data that supports the conclusions. Experiments must have been conducted rigorously, with appropriate controls, replication, and sample sizes. The conclusions must be drawn appropriately based on the data presented.yes

7. PLOS authors have the option to publish the peer review history of their article (what does this mean? ). If published, this will include your full peer review and any attached files.

**Do you want your identity to be public for this peer review?** For information about this choice, including consent withdrawal, please see our Privacy Policy .

Reviewer #1: No

Reviewer #2: **Yes: ** Linlu Dong

---

## [Author Response · Author response to Decision Letter 2]

20 Nov 2024

Dear Reviewers,

Thank you very much for your valuable and insightful feedback on our manuscript. We greatly appreciate the time and effort you have dedicated to providing your comments and suggestions. In response, we have carefully revised the manuscript to address all your concerns and ensure clarity, consistency, and accuracy throughout.

We have attached a detailed point-by-point response, outlining how each of your comments has been addressed. If our revisions do not fully meet your expectations, we kindly request another opportunity to further improve the manuscript until it satisfies your requirements.

Once again, thank you for your thoughtful review. Your guidance has been instrumental in enhancing the quality of our work, and we are sincerely grateful for your support.

Best regards,

Yours sincerely,

Linna Ji

---

## [Editor Report · Decision Letter 2]

25 Nov 2024

MAF: An algorithm based on multi-agent characteristics for infrared and visible video fusion

PONE-D-24-28516R2

Dear Dr. Ji,

We’re pleased to inform you that your manuscript has been judged scientifically suitable for publication and will be formally accepted for publication once it meets all outstanding technical requirements.

Kind regards,

Yawen Lu, Ph.D

Academic Editor

PLOS ONE

Additional Editor Comments (optional):

Dear authors:

Regarding your revised submission PONE-D-24-28516R2:

MAF: An algorithm based on multi-agent characteristics for infrared and visible video fusion

We have checked your revision and are announcing that your work has been Accepted for publication in PLOS ONE.

Please follow the following steps and provide a camera-ready version of your manuscript. Congratulation again!
---

## [Editor Report · Acceptance letter]

PONE-D-24-28516R2

PLOS ONE

Dear Dr. Ji,

I'm pleased to inform you that your manuscript has been deemed suitable for publication in PLOS ONE. Congratulations! Your manuscript is now being handed over to our production team.

Kind regards,

on behalf of

Dr. Yawen Lu

Academic Editor

PLOS ONE